# An individual's physique is associated with the length of the reconstruction route via the retrosternal approach

**Kotaro Honda[1], Sang-Woong Lee**[1]***, Masaru Kawai[2], Ryo Tanaka[1], Yoshiro Imai[1], Kentaro Matsuo[1], Kazuhisa Uchiyama[1]**

**1** Department of General and Gastroenterological Surgery, Osaka Medical and Pharmaceutical University, Osaka, Japan, **2** Department of Gastroenterological Surgery, Hirakata City Hospital, Osaka, Japan

* sang-woong.lee@ompu.ac.jp

**Data Availability Statement:** All relevant data are within the paper and its Supporting Information files.

## Abstract

We hypothesized that an individual's physique is related to reconstruction length, as organs reconstructed via the retrosternal route are curved toward the ventral side. This study aimed to determine factors contributing to the reconstruction length of the retrosternal route. Fifty patients underwent subtotal esophagectomy with esophagogastric reconstruction via the posterior mediastinal route between 2010 and 2014; the esophagus–stomach and posterior mediastinal route lengths were measured to evaluate whether they could be approximated. Forty patients underwent reconstruction via the retrosternal route between 2015 and 2020; the esophagus–stomach and retrosternal route lengths were compared, and contributing factors were analyzed. Each length was measured perioperatively using three-dimensional enhanced computed tomography images. The associated factors obtained included age, sex, height, body weight, body mass index, thickness and height of the thorax, depth of the thoracic inlet space, thoracic curve, left hepatic lobe volume, and the thickness and height of the liver. The length of the esophagus–stomach could approximate that of the posterior mediastinal route [posterior mediastinal-esophagus–stomach; 0.04 (-0.5–0.6) cm, $p = 0.77$]. Using three-dimensional enhanced computed tomography images, the lengths of the esophagus–stomach and retrosternal routes were comparable, despite variability [retrosternal-esophagus–stomach; 0.72 (-0.1–1.8) cm, $p = 0.095$]. Analyzing factors associated with the length revealed a positive correlation of body weight, body mass index, and thickness of the thorax with the difference. A higher body mass index (OR = 1.7, 95% CI 1.1–2.8, $p = 0.007$) was associated with a longer retrosternal route in the multivariate analysis. An individual's physique is associated with the reconstruction length; particularly, the length of the retrosternal route was longer in patients with a high body mass index.

## Introduction

Subtotal esophagectomy, the typical treatment for esophageal cancer, remains a highly invasive surgical procedure with various complications, including anastomotic leakage or ischemia of the reconstructed organs, such as the stomach, jejunum, or ileocolon [1,2]. Specifically,

**Funding:** The author received no specific funding for this work.

**Competing interests:** The authors have declared that no competing interests exist.

anastomotic leakage caused by both surgical [3–8] and patient factors [9–12] increases mortality [13] and worsens patient prognosis [14]. Among surgical factors, a strain on the anastomotic site and low blood flow of the reconstructed organs are likely to cause anastomotic leakage and ischemia; therefore, the length of the reconstruction route is thought to be associated with an anastomosis-related complication. One strategy to prevent anastomosis-related complications can be to pull the reconstructed organ up via the shortest route. Especially in cases of low blood flow or when the reconstructed organ is too short, the decision on the shortest route should be made to avoid postoperative adverse events.

The posterior mediastinal (PM) and retrosternal (RS) routes are the standard reconstruction routes following radical esophagectomy for cancer in 40.2% and 40.1% of cases, respectively, as reported by the Comprehensive Registry of Esophageal Cancer in Japan in 2013 [15]. The reconstruction route and organs are selected after considering the safety issues and postoperative quality of life, and the preferred surgical procedure depends on the individual institute or surgeon. Although several studies have explored the shortest reconstruction route after subtotal esophagectomy, there remains a lack of consensus on the optimal route for the reconstruction [16–21].

We hypothesized that an individual's physique is associated with the length of the reconstruction route via the RS route curved toward the ventral side, and the length of the reconstruction route varies with each case. Thus, this study aimed to analyze the factors that contribute to the reconstruction length of the RS route following subtotal esophagectomy.

## Materials and methods

### Ethical information

This study was approved by the Institutional Review Board of Osaka Medical and Pharmaceutical University and conformed to the provisions of the Declaration of Helsinki in 1995 (as revised in Edinburgh 2000). The requirement for informed consent from the patients was waived by the review board because of the retrospective study design.

### Patients

At our department, thoracoscopic subtotal esophagectomy with three-field lymph node dissection in the prone position is the standard treatment approach for thoracic esophageal cancer. Reconstruction is performed following the modified Collard [22,23] or triangulating stapled cervical esophagogastric anastomosis [24], using the narrow gastric conduit constructed under laparoscopic assistance. The anastomosis is performed at the lesser curvature of the gastric conduit and as much on the anal side as possible, where the blood flow is optimal. The RS route is created by tunneling the dorsal sternum to the neck after a small laparotomy, and then pulling the gastric conduit up to the neck. The manubrium or clavicle is not resected.

Among consecutive patients who underwent the reconstruction via the PM route following subtotal esophagectomy between 2010 and 2014, the esophagus–stomach (ES) length was retrospectively compared with the length of the PM route to evaluate whether they were approximate in patients who underwent reconstruction via the PM route. The length of the ES compared with that of the PM route was defined as the length of the esophagus plus the distance from the esophageal hiatus to the origin of the gastroduodenal artery (GDA).

Subsequently, consecutive patients who underwent the reconstruction via the RS route following a subtotal esophagectomy between 2015 and 2020 were examined. From 2015 onwards, we used the RS route as the standard reconstruction route to avoid esophagobronchial fistula formation. The lengths of the ES and RS routes were measured as alternatives to the lengths of

the PM and RS routes, and factors associated with the length of the reconstruction route were analyzed in all the patients.

Contrast-enhanced computed tomography (CT; slice thickness, $\leq 2.0$ mm) was carried out before and after surgery in all patients. The Kocher maneuver was not performed routinely during the esophagectomy; however, the patients on whom this was performed were excluded, as the measurement points may have shifted. No patients had hiatal hernia of the esophagus.

### Measurements

Reconstruction lengths were measured by three-dimensional contrast-enhanced CT (3D-CT) using the Volume Analyzer SYNAPSE VINCENT by Fujifilm (Japan). All scans were obtained in a neutral position during inspiration. Each length was measured by plotting the established reference points of the intestinal lumen from the lower border of the cricoid cartilage to the origin of the GDA. The lengths of the ES and PM routes were measured by plotting the reference points at the lower border of the cricoid cartilage, upper border of the jugular notch, lower border of the tracheal bifurcation, middle point between the lower border of the tracheal bifurcation and esophageal hiatus, esophageal hiatus, and origin of the GDA (Fig 1A). The reconstruction length of the RS route was measured by plotting reference points at the lower border of the cricoid cartilage, upper border of the jugular notch, lower border of the tracheal bifurcation, lower border of the body of the sternum, lower border of the left lobe of the liver, and origin of the GDA (Fig 1B).

To analyze the factors contributing to the reconstruction length of the RS route, factors related to an individual's physique and the difference between the lengths of the ES and RS routes were investigated; patients with longer and shorter routes were divided into subgroups within the RS group and were compared by each parameter. All the following factors obtained through 3D-CT in the mid-sagittal central plane were measured: thickness and height of the thorax, depth of the thoracic inlet space, thoracic curve, left hepatic lobe volume, and thickness and height of the liver (Fig 2). The patients' age, sex, height, body weight, and body mass index (BMI) were also obtained. The measurements were carried out in triplicate, and the mean was used as the measured value.

### Statistical analysis

Values are presented as medians (ranges). Continuous data were compared using the Wilcoxon rank-sum test, and the differences in each group were analyzed using Fisher's exact probability test. The relationship between the length of the reconstruction route and the measured factors was tested using Spearman's rank correlation coefficient. A multivariate logistic regression was performed to analyze the factors contributing to the reconstruction length. The receiver operating characteristic analysis was performed to calculate the area under the receiver operating characteristic curve and the cut-off value. All analyses were performed using JMP® 14 (SAS Institute Inc., Cary, NC, USA), yielding two-sided $p$-values; $p < 0.05$ was considered significant.

### Results

Table 1 shows the characteristics of the patients and tumors in each group by the reconstruction routes. A total of 50 consecutive patients underwent reconstruction via the PM route following subtotal esophagectomy between 2010 and 2014, and 40 consecutive patients underwent reconstruction via the RS route following subtotal esophagectomy between 2015 and 2020. No significant difference in anastomosis-related complications was found between the two groups.

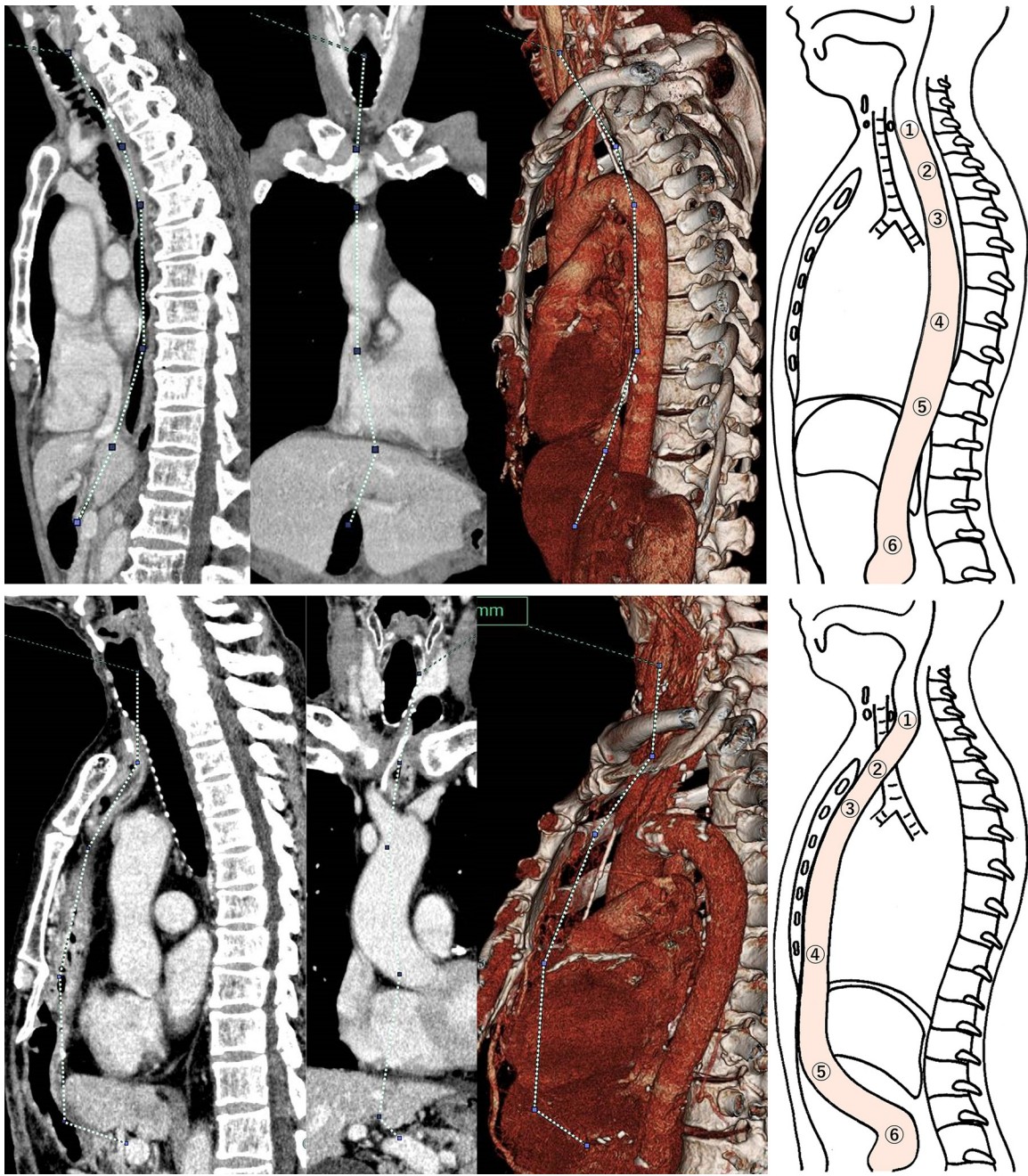

**Fig 1.** (a) Reference points for measuring the length of the esophagus and stomach and the posterior mediastinal route. ① Cricoid cartilage, ② jugular notch, ③ tracheal bifurcation, ④ middle point between ③ and ⑤, ⑤ esophageal hiatus, and ⑥ gastroduodenal artery. (b) Reference points for measuring the length of the retrosternal route. ① Cricoid cartilage, ② jugular notch, ③ tracheal bifurcation, ④ lower border of the body of sternum, ⑤ lower border of the left hepatic lobe, and ⑥ gastroduodenal artery.

Table 2 shows the lengths of the routes; in PM reconstructions (n = 50), the median lengths of the ES and PM routes were 33.0 (32.0–34.2) and 33.1 (31.7–33.8) cm, respectively. The difference between the ES and PM routes (PM–ES) was 0.04 (-0.5–0.6) cm ($p = 0.77$). In the RS reconstructions (n = 40), the median lengths of the ES and RS routes were 33.3 (32.0–34.7) and 34.0 (33.2–35.3) cm, respectively. The median difference between the ES and RS route

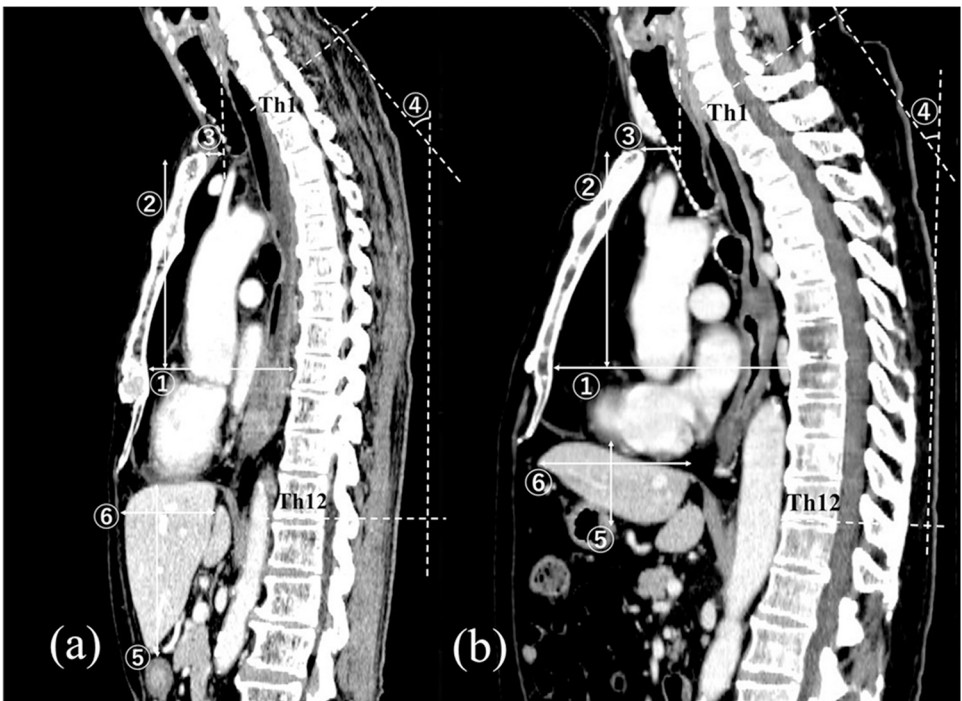

**Fig 2. Measurement methods for factors contributing to the length of the reconstruction route.** (a) Thin thorax: BMI of 17.9 kg/m$^2$, thoracic thickness of 9.7 cm, and RS–ES length of -0.7 cm. (b) Thick thorax: BMI of 26.3 kg/m$^2$, thoracic thickness of 14.3 cm, and RS–ES length of 1.8 cm. Each point was measured in the mid-sagittal central plane. ① The thickness of the thorax was measured from the lower border of the body of the sternum to the vertebral body. ② The height of the thorax was measured from the upper border of the jugular notch to the lower border of the body of the sternum. ③ The depth of the thoracic inlet space was measured from the upper border of the jugular notch to the lower border of the cricoid cartilage. ④ The thoracic curve was measured at the angle of the extended intersection between Th1 upper and Th12 lower borders of the vertebral body. ⑤ The height of the liver was measured at the cranial–caudal distance. ⑥ The thickness of the liver was measured at the anterior–posterior distance.

(RS–ES) was 0.72 (-0.1–1.8) cm; therefore, the RS route was slightly longer than the ES route, although no significant difference was noted ($p$ = 0.095). The RS $\geq$ ES group included 30 cases, whereas the RS < ES group included 10 cases; meanwhile, the anastomosis-related complications in each group comprised six cases and one case, respectively, in which there was no significant difference ($p$ = 0.66).

In addition, the factors contributing to the length of the reconstruction route were investigated. Spearman's rank correlation analysis showed that the difference between the lengths of the ES and RS routes was positively correlated with body weight ($\rho$ = 0.42, $p$ = 0.007), BMI ($\rho$ = 0.52, $p$ = 0.001), and thickness of the thorax ($\rho$ = 0.39, $p$ = 0.01), as shown in Fig 3. The univariate analysis indicated that the length of the RS route was significantly associated with BMI ($p$ = 0.006) and the height of the liver ($p$ = 0.04). Furthermore, in the multivariate analysis, only BMI (OR = 1.7, 95% CI 1.1–2.8, $p$ = 0.007) was associated with the length of the RS route, as shown in Table 3. In this study, the cut-off value of BMI was 19.6 kg/m$^2$ (area under curve 0.8). There were five obese cases with BMI >25 kg/m$^2$, in which the RS route was longer than the ES route and no anastomosis-related complications were observed.

## Discussion

This study has some notable findings. First, comparison of the reconstruction lengths of the ES and PM routes revealed that the preoperative length of the PM route could approximate

**Table 1. Characteristics of each reconstruction route group.**

|  | PM[a] (n = 50) | RS[b] (n = 40) | *p*-value |
|---|---|---|---|
| Age, years, median (range) | 69 (64–73) | 70.5 (65–75) | 0.29 |
| Sex |  |  |  |
| Male/Female | 46/4 | 29/11 | 0.02 |
| Height, cm, median (range) | 164.5 (161.8–169.0) | 164.0 (159.5–168.8) | 0.29 |
| Body weight, cm, median (range) | 58.0 (51.0–63.4) | 54.6 (45.0–61.9) | 0.05 |
| BMI, kg/m$^2$, median (range) | 21.5 (19.2–23.3) | 19.6 (17.6–22.7) | 0.05 |
| Histology |  |  |  |
| Squamous cell carcinoma | 47 (94) | 39 (97.5) | 0.49 |
| Adenocarcinoma | 2 (4) | 0 (0) |  |
| Others | 1 (2) | 1 (2.5) |  |
| Tumor location |  |  |  |
| Upper/middle/lower | 5(10) / 32(64) / 13(26) | 3(7.5) / 23(57.5) / 14(35) | 0.81 |
| pStage[c] |  |  |  |
| I | 25 (50) | 7 (17.5) | 0.01 |
| II | 4 (8) | 8 (20) |  |
| III | 16 (32) | 17 (42.5) |  |
| IV | 5 (10) | 8 (20) |  |
| Neoadjuvant therapy |  |  |  |
| No | 31 (62) | 9 (22.5) | 0.001 |
| Chemotherapy | 17 (34) | 27 (67.5) |  |
| Chemoradiotherapy | 2 (4) | 4 (10) |  |
| Anastomotic complications |  |  |  |
| Anastomotic leak | 10 (20) | 7 (17.5) | 0.79 |
| Ischemia | 1 (2) | 2 (5) | 0.58 |

Data are presented as n (%), unless otherwise indicated.

[a]Posterior mediastinal

[b]Retrosternal

[c]TNM classification (UICC 8th edition).

that of the ES route. Second, comparison of the lengths of the ES and RS routes in patients who underwent reconstruction via the RS route showed that the ES and RS reconstruction routes had similar lengths, though the difference between the ES and RS routes varied in each

**Table 2. Length of each route and difference between post- and preoperative lengths.**

|  | Preoperative length (ES[a]) | Postoperative length (PM[b]/RS[c]) | Difference (Post-Pre) | *p*-value |
|---|---|---|---|---|
| PM (n = 50) | 33.0 (32.0–34.2) | 33.1 (31.7–33.8) | 0.04 (-0.5–0.6)[d] | 0.77 |
| RS (n = 40) | 33.3 (32.0–34.7) | 34.0 (33.2–35.3) | 0.72 (-0.1–1.8)[e] | 0.095 |

Values are expressed as median distance and range.

[a]Esophagus–stomach

[b]Posterior mediastinal

[c]Retrosternal.

[d]Difference between the lengths of the ES and PM routes (PM–ES). A smaller PM–ES value indicates that the lengths of the ES and PM routes are comparable.

[e]Difference between the lengths of the ES and RS routes (RS–ES). A positive RS–ES value indicates that the RS route is longer than the PM route, as the length of the ES can be approximated to that of the PM route.

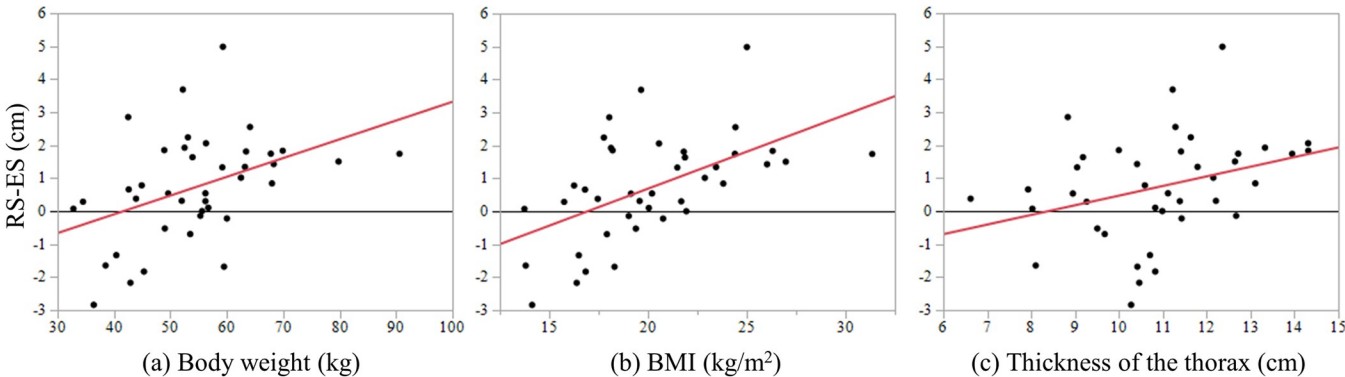

**Fig 3. Relationships between the three factors and differences in reconstruction lengths.** The three factors associated with the difference between the length of the retrosternal route and that of the esophagus–stomach (RS–ES) were analyzed using Spearman's rank correlation coefficient; positive correlations were observed with each factor. **(a)** Body weight: $\rho = 0.42$, $p = 0.007$; **(b)** BMI: $\rho = 0.52$, $p = 0.001$; **(c)** Thickness of the thorax: $\rho = 0.39$, $p = 0.01$.

case (Table 2). Finally, analysis of the factors contributing to the length of the reconstruction route revealed that the RS route was longer in patients with a high BMI (Fig 3 and Table 3). Clinicians could consider using the Kocher maneuver and creating a long gastric conduit when treating obese patients. To the best of our knowledge, this study was the first report on the reconstruction length following subtotal esophagectomy measured by perioperative 3D-CT.

Table 4 lists seven studies comparing the differences between the PM and RS routes, including the present study. Of them, except for our study, four studies utilized cadaver's body and the remaining two directly performed measurements during surgery. The proximal reference

**Table 3. Relationship between factors and reconstruction routes.**

| Variables | RS[a] ≧ ES[b] (n = 30) | RS < ES (n = 10) | Univariate analysis p-value | Multivariate analysis | |
| --- | --- | --- | --- | --- | --- |
| | | | | OR (95% CI) | p-value |
| Age | 70.5 (64.8–77.0) | 69.5 (63.8–73.0) | 0.38 | | |
| Sex (male/female) | 22 / 8 | 7 / 3 | 1.00 | | |
| Height (cm) | 163.5 (159.2–167.2) | 165.5 (160.1–171.1) | 0.37 | | |
| Body weight (kg) | 56.2 (49.4–63.6) | 47.2 (39.9–56.4) | 0.05 | | |
| BMI (kg/m²) | 21.0 (18.1–24.0) | 17.4 (15.8–19.1) | 0.006 | 1.7 (1.1–2.8) | 0.007 |
| Thickness of the thorax (cm) | 11.3 (9.2–12.4) | 10.4 (9.6–11.0) | 0.25 | 0.6 (0.2–1.4) | 0.24 |
| Height of the thorax (cm) | 13.9 (13.2–15.6) | 14.9 (13.0–15.8) | 0.84 | | |
| Depth of thoracic inlet space (cm) | 2.7 (1.6–3.1) | 2.5 (1.1–3.4) | 0.56 | | |
| Thoracic curve (°) | 25.0 (20.1–33.4) | 27.5 (23.8–37.6) | 0.27 | 1.0 (0.82–1.1) | 0.6 |
| Left hepatic lobe volume (ml) | 198.8 (155.0–238.0) | 231.3 (173.5–263.4) | 0.30 | | |
| Height of the liver (cm) | 5.9 (4.6–7.8) | 8.0 (6.2–8.6) | 0.04 | 0.6 (0.2–1.0) | 0.07 |
| Thickness of the liver (cm) | 7.5 (6.1–8.5) | 6.3 (5.8–7.1) | 0.13 | | |

[a]Retrosternal

[b]Esophagus–stomach, assumed to be the same length as the posterior mediastinal route.

**Table 4. Studies on the differences between RS[a] and PM[b].**

| Authors | n | Subjects | Proximal reference point | Distal reference point and RS-PM value (cm) | | | |
|---|---|---|---|---|---|---|---|
| | | | | Celiac axis | GDA[c] | Pyloric ring | Duodenum[d] |
| Orringer and Sloan [16] | 10 | Cadaver | Unknown | Unknown[e] | | | |
| Ngan and Wong [17] | 20 | Cadaver | Cricoid cartilage | 1.9 | | | |
| Coral et al. [18] | 50 | Cadaver | Cricoid cartilage | 5.3 | 2.5 | | |
| Chen et al. [19] | 60 | Patient in surgery | Cricoid cartilage | | | -2.8 | |
| Hu et al. [20] | 20 | Cadaver | Cricoid cartilage | 2.5 | -1.3 | -1.5 | |
| Yasuda et al. [21] | 112 | Patient in surgery | Cricoid cartilage | | | | -2.3 |
| Current study | 40 | Patient | Cricoid cartilage | | 0.7 | | |

[a]Retrosternal

[b]Posterior mediastinal

[c]Gastroduodenal artery.

[d]Superior border of the duodenum arising from the head of the pancreas.

[e]Orringer and Sloan reported that the RS route was longer than the PM route; however, data were not available.

point was the lower border of the cricoid cartilage, similar to that reported in previous studies. In all cases, the level of the anastomosis was accurately determined via the CT images because a stapling device was used for cervical anastomosis. Additionally, the level of anastomosis in all cases was determined at the oral side rather than at the jugular notch. Although the proximal reference point for measurement was the lower border of the cricoid cartilage and not the level of anastomosis, the difference did not affect the measurement results. The distal reference points varied in each study, including the GDA, celiac axis, pyloric ring, and duodenum at the level of the pancreatic head. In our study, the origin of the GDA was measured, as the point fixed in the retroperitoneum was easily identified on CT images and did not shift during surgery. Some previous studies have reported that the PM route is the shortest [16–18], whereas others have reported that the RS route is the shortest [19–21]. Given these contrasting results, there remains a lack of consensus on the shortest route. Moreover, no studies have investigated the relation of the reconstructed length with patients' characteristics. In our study, we found no differences in the length of the PM and RS routes, though the differences were significant in some cases. An analysis of some of the patient's demographic characteristics revealed that the BMI contributed to the reconstruction length. To the best of our knowledge, no study has so far showed the positive correlation between the reconstruction length and an individual's physique, including BMI, body weight, and thickness of the thorax (Fig 3). Previous studies analyzing actual lengths measured during surgery could reveal reliable findings; however, in a practical situation, surgeons often like to know the information preoperatively in each case. In the present study, the results based on measurements using 3D-CT images did not show any significant difference in length between the RS and PM routes; however, we found that the reconstruction length was related to physical factors. The variability of the results in previous reports may be attributed to the physique of the enrolled patients, which should be considered when comparing reconstruction lengths. The reconstructed organs were curved regardless of the reconstruction route; therefore, the 3D measurement method used in the present study was considered a novel approach that enabled preoperative measurement of the curved length, based on the individual's physique, and helped predict a shorter reconstruction route.

Several studies have examined the relationship between the reconstruction routes and anastomotic leakage, where no significant difference was observed in the incidence of anastomotic leakage between the RS and PM routes [6,25–30]. The length is not the only cause of complications; anastomotic leakage is caused by a combination of several factors [3–12]. However, the reconstruction length is considered a critical factor related to anastomotic leakage due to an increase in the incidence of similar complications if the reconstructed organs are anastomosed at a site with poor blood flow. The anastomotic leakage in esophagectomy with cervical anastomosis in this study occurred in 17 (18.9%) of the 90 total cases, including 10 (20%) of the 50 cases wherein the PM route was employed and seven (17.5%) of the 40 cases wherein the RS route was used. Notably, the incidence of anastomotic leakage in our study was higher than that reported in other studies. While our results offer some insight on the subject, further study is needed to identify the optimal reconstructive route to reduce anastomotic leakage. Although our series did not reveal a relationship between the length of the reconstruction route and anastomotic leakage and did not reveal a relationship between the BMI and anastomotic leakage, a study reported that obesity is associated with anastomotic leakage in esophagectomy [31]. In Japan, the selection of the reconstruction route depends on the surgeon and institute, and the PM and RS routes are standard reconstruction routes employed after radical esophagectomy for cancer in 40.2% and 40.1% of cases, respectively, with no significant differences reported between the surgical outcomes of the two routes. At our institute, the RS route is the standard reconstruction route undertaken when serious complications, such as esophagobronchial fistula, are likely to develop after reconstruction using the PM route. Our study corroborates the results of previous studies in Japan by showing that the surgical outcomes of esophagectomy do not differ between both routes. The results of our study suggest that the PM route should be selected to avoid anastomosis-related complications when treating obese patients with a high perioperative risk. In this study, five overweight cases had BMIs >25 kg/$m^2$, including one with BMI >30 kg/$m^2$; in all these cases, the RS route was longer than the PM route. Thus, even if the standard route for reconstruction is RS, we recommend using the PM route when treating obese patients (BMI >25 kg/$m^2$) to avoid anastomosis-related complications and potentially improve the surgical outcome of esophagectomy for esophageal cancer in such patients. To verify our results, it is necessary to further investigate the relationship in more obese patients with esophageal cancer.

This study has a few limitations. First, the study followed a retrospective design, and the results may have a patient selection bias. Second, the time point for surgeries varied owing to factors such as equipment and surgeon availability. This could also introduce variations in the outcomes of esophagectomies. Third, there may be a discrepancy between the actual length measured during the surgery and the length measured using CT. In this study, both these lengths were not compared. However, the study findings revealed that the reconstruction length was clearly related to the individual's physique. Elucidation of the factors contributing to the reconstruction length and comparison of the differences in the length of the PM and RS routes before esophagectomy using the 3D measurement method can help clinicians select the optimal reconstruction route. Consequently, the outcome of esophagectomies for esophageal cancer may improve. Further studies exploring the relationships between reconstruction lengths, individual's physique, and outcomes of esophagectomies are needed to confirm these findings.

## Conclusion

The length of the reconstruction route was found to be dependent on an individual's physique; in particular, the RS route was longer in patients with a high BMI. Thus, when treating obese

patients with a high perioperative risk, clinicians could consider using the Kocher maneuver to create a long gastric conduit and employing the PM route to potentially reduce the incidence of anastomosis-related complications.

## Supporting information

**S1 Data.**
(XLSX)

**S2 Data.**
(XLSX)

## Author Contributions

**Conceptualization:** Kotaro Honda.

**Data curation:** Kotaro Honda, Masaru Kawai, Ryo Tanaka, Yoshiro Imai, Kentaro Matsuo.

**Formal analysis:** Kotaro Honda.

**Investigation:** Kotaro Honda.

**Methodology:** Kotaro Honda.

**Supervision:** Sang-Woong Lee, Kazuhisa Uchiyama.

**Validation:** Kotaro Honda.

**Writing – original draft:** Kotaro Honda.

**Writing – review & editing:** Sang-Woong Lee.

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
