## [Decision Letter · Decision Letter 0]

11 Feb 2022

PONE-D-21-33230An individual’s physique affects the length of the reconstruction route via the retrosternal approachPLOS ONE

Dear Dr. Lee,

Thank you for submitting your manuscript to PLOS ONE. After careful consideration, we feel that it has merit but does not fully meet PLOS ONE’s publication criteria as it currently stands. Therefore, we invite you to submit a revised version of the manuscript that addresses the points raised during the review process.

Please bare in mind that substantial issues have been raised by the reviewers. Therefore a major revision of the article is mandatory before resubmitting, addressing all issues raised. 

We look forward to receiving your revised manuscript.

Kind regards,

Eric D. Roessner, Prof.

Academic Editor

PLOS ONE

Journal Requirements:

Reviewers' comments:

Reviewer's Responses to Questions

**Comments to the Author**

1. Is the manuscript technically sound, and do the data support the conclusions?

Reviewer #1: Yes

Reviewer #2: Yes

2. Has the statistical analysis been performed appropriately and rigorously? 

Reviewer #1: I Don't Know

Reviewer #2: Yes

3. Have the authors made all data underlying the findings in their manuscript fully available?

Reviewer #1: Yes

Reviewer #2: No

4. Is the manuscript presented in an intelligible fashion and written in standard English?

Reviewer #1: Yes

Reviewer #2: Yes

5. Review Comments to the Author

Reviewer #1: I congratulate Dr. Sang-Woong Lee and his colleagues on a clear and well-written manuscript.

I have several comments and questions:

1) The findings from this article are not new, although the method is more applicable to clinical practice.

2) Please specify if you included patients with hiatal hernia. My assumption is that you had no patients with hiatal hernia based on pathology and BMI, but it is important to specify this.

3) Presumably, your practice changed in 2015; it would appear that you switched to performing mainly RS reconstructions. Otherwise, why the 2 different time periods? Why did you switch?

4) The leak rate is high when compared to other reports in the literature; please explain how your leak rate compared with that of other series of esophagectomy with cervical anastomosis.

5) Your study would carry much more weight and validation to your measurement method if you would also provide a measurement of the anastomosis from the teeth as determined by endoscopy. How do we know the level of the anastomosis is the same? That also significantly affects how we interpret these data.

6) Your technique for retrosternal reconstruction should be explained in the methods. Where exactly do you perform the anastomosis? Do you remove a portion of the manubrium or the sterno-clavicular joint?

7) Conclusion: What should a practicing esophageal surgeon do with this information? Outcomes were the same. What is the practical relevance of this study?

8) As a North American surgeon, I have learned from your article that primary reconstruction via the retrosternal route is a feasible approach. In North America, we reserve the retrosternal rout mainly for delayed reconstruction.

Reviewer #2: Overall, the aim of the study is clear, the results are as well. Nevertheless, a few things remain unclear:

A. The results state, that retrosternal approach might be associated to higher anastomotic leakage in higher BMI patients due to a longer conduit.

B. Clavicular or sternal resection is often needed to avoid postoperative dysphagia in patients with retrosternal approach. This is congruent to our experience. Therefore our clinic has set the posterior mediastinal approach as the standard procedure.

 In which case would you still recommend retrosternal approach and therefore prove relevance of these findings?

In your manuscript you mentioned that higher BMI is associated to longer routes for RS routes. Fig 3 shows some kind of visualization. Nevertheless, for clinical use, a consequence of these findings could be stated. Is there possibly a cut-off value for significant lengthening of the route? Is there a recommendation for which approach to prioritize or avoid with a certain BMI?

The discussion lacks a conclusion for clinicians towards which consequence your findings might have (literature states that anastomotic leakage rates overall are not higher in the retrosternal approach than in the posterior mediastinal approach, as you stated earlier in your manuscript).

6. PLOS authors have the option to publish the peer review history of their article (what does this mean?). If published, this will include your full peer review and any attached files.

Reviewer #1: No

Reviewer #2: No

---

## [Author Response · Author response to Decision Letter 0]

25 Mar 2022

Responses to Editor’s and Reviewers’ Comments

To Reviewer #1

Reviewer’s comment 1

The findings from this article are not new, although the method is more applicable to clinical practice.

Response to comment 1

Thank you for your comment. We believe that the 3D measurement method used in the present study was a novel approach to enable preoperative measurement of the curved length, based on the individual’s physique, and predict a shorter reconstruction route. We found that the retrosternal route (RS) was longer in patients with a high BMI, and thus suggest practicing esophageal surgeons to consider using the Kocher maneuver for creating a long gastric conduit when treating obese patients. It is necessary to further investigate the relationship between reconstruction length and surgical outcomes of esophagectomy in patients with a high BMI. We have added the following sentence to the Discussion section: “...therefore, the 3D measurement method used in the present study was considered a novel approach that enabled preoperative measurement of the curved length, based on the individual’s physique, and helped predict a shorter reconstruction route.” (Page 14, lines 256–258) 

Reviewer’s comment 2

Please specify if you included patients with hiatal hernia. My assumption is that you had no patients with hiatal hernia based on pathology and BMI, but it is important to specify this.

Response to comment 2

Thank you for your comment. In this study, each length was measured by plotting the established reference points from the lower border of the cricoid cartilage to the origin of the GDA, including the esophageal hiatus. Therefore, we believe that the presence of a hiatal hernia of the esophagus did not affect reconstruction length. Nonetheless, we have added the following sentence to the Materials and Methods section: “No patients had hiatal hernia of the esophagus.” (Page 5, lines 100).

Reviewer’s comment 3

Presumably, your practice changed in 2015; it would appear that you switched to performing mainly RS reconstructions. Otherwise, why the 2 different time periods? Why did you switch?

Response to comment 3

Thank you for your comment. At our department, the RS route has been used as the standard reconstruction route since 2015 for cases wherein serious complications, such as esophagobronchial fistula, may develop after reconstruction using the posterior mediastinal (PM) route. According to your suggestion, we have added the following sentence to the Patients section: “From 2015 onwards, we used the RS route as the standard reconstruction route to avoid esophagobronchial fistula formation.” (Page 5, lines 92–94)

Reviewer’s comment 4

The leak rate is high when compared to other reports in the literature; please explain how your leak rate compared with that of other series of esophagectomy with cervical anastomosis.

Response to comment 4

Thank you for your pertinent comment. The anastomotic leakage in esophagectomy with cervical anastomosis in this study occurred in 17 (18.9%) of the 90 total cases, including in 10 (20%) of 50 cases wherein the PM route was used and seven (17.5%) of the 40 cases wherein the RS route was used. The reported leak rates of esophagectomy with cervical anastomosis are 12.3 % [1] and 13.6 % [27], and as you pointed out, the leak rate in our study is higher than these reported values. In our department, thoracoscopic subtotal esophagectomy was considered the standard treatment approach for thoracic esophageal cancer since 2010, but the standard reconstruction route was changed from the PM route to the RS route from 2015 onwards. Both routes (PM and RS) result in a high incidence of anastomotic leakage during the induction phase. Thus, the search is ongoing for optimal reconstructive routes that reduce anastomotic leakage, and the present study is one of the attempts to address this knowledge gap. In response to your suggestion, we have added the following text to the Discussion section: “The anastomotic leakage in esophagectomy with cervical anastomosis in this study occurred in 17 (18.9%) of the 90 total cases, including 10 (20%) of the 50 cases wherein the PM route was employed and seven (17.5%) of the 40 cases wherein the RS route was used. Notably, the incidence of anastomotic leakage in our study was higher than that reported in other studies. While our results offer some insight on the subject, further study is needed to identify the optimal reconstructive route to reduce anastomotic leakage.” (Pages 15 & 16, lines 272–277)

Reviewer’s comment 5

Your study would carry much more weight and validation to your measurement method if you would also provide a measurement of the anastomosis from the teeth as determined by endoscopy. How do we know the level of the anastomosis is the same? That also significantly affects how we interpret these data.

Response to comment 5

Thank you for your comment. In this study, we could not measure the anastomosis with reference to the teeth using endoscopy. However, in response to your suggestion, we have added the following sentence to the Discussion section: “The proximal reference point was the lower border of the cricoid cartilage, similar to that reported in previous studies. In all cases, the level of the anastomosis was accurately determined via the CT images because a stapling device was used for cervical anastomosis. Additionally, the level of anastomosis in all cases was determined at the oral side rather than at the jugular notch. Although the proximal reference point for measurement was the lower border of the cricoid cartilage and not the level of anastomosis, the difference did not affect the measurement results.” (Page 13, lines 228–234)

Reviewer’s comment 6

Your technique for retrosternal reconstruction should be explained in the methods. Where exactly do you perform the anastomosis? Do you remove a portion of the manubrium or the sterno-clavicular joint?

Response to comment 6

Thank you for your comment. According to your suggestion, we have added the following sentence to the Material and Methods section: “The anastomosis is performed at the lesser curvature of the gastric conduit and as much on the anal side as possible, where the blood flow is optimal. The RS route is created by tunneling the dorsal sternum to the neck after a small laparotomy, and then pulling the gastric conduit up to the neck. The manubrium or clavicle is not resected.” (Page 4, lines 81–84) 

Reviewer’s comment 7

Conclusion: What should a practicing esophageal surgeon do with this information? Outcomes were the same. What is the practical relevance of this study?

Response to comment 7

Thank you for your suggestion. This study found that the RS route was longer in patients with a high BMI. It is necessary to further investigate the relationship between reconstruction length and surgical outcomes of esophagectomy in patients with a high BMI. However, the results of this study indicate that practicing esophageal surgeons could consider using the Kocher maneuver to create a long gastric conduit when treating obese patients. According to your suggestion, we have added the following sentence to the Conclusion: “Thus, when treating obese patients with a high perioperative risk, clinicians could consider using the Kocher maneuver to create a long gastric conduit and employing the PM route to potentially reduce the incidence of anastomosis-related complications.” (Page 17, lines 312–313).

Reviewer’s comment 8

As a North American surgeon, I have learned from your article that primary reconstruction via the retrosternal route is a feasible approach. In North America, we reserve the retrosternal rout mainly for delayed reconstruction.

Response to comment 8

Thank you for your advice. In Japan, the PM and RS routes are the standard reconstruction routes following radical esophagectomy for cancer in 40.2% and 40.1% of cases, respectively. Further, the selection of the reconstruction route depends on the surgeon and institute. Existing literature shows no significant difference in the surgical outcomes of esophagectomy between both routes. According to your suggestion, we have added the following sentence to the Discussion section: “In Japan, the selection of the reconstruction route depends on the surgeon and institute, and the PM and RS routes are standard reconstruction routes employed after radical esophagectomy for cancer in 40.2% and 40.1% of cases, respectively, with no significant differences reported between the surgical outcomes of the two routes. At our institute, the RS route is the standard reconstruction route undertaken when serious complications, such as esophagobronchial fistula, are likely to develop after reconstruction using the PM route. Our study corroborates the results of previous studies in Japan by showing that the surgical outcomes of esophagectomy do not differ between both routes. The results of our study suggest that the PM route should be selected to avoid anastomosis-related complications when treating obese patients with a high perioperative risk. In this study, five overweight cases had BMIs >25 kg/m2, including one with BMI >30 kg/m2; in all these cases, the RS route was longer than the PM route. Thus, even if the standard route for reconstruction is RS, we recommend using the PM route when treating obese patients (BMI >25 kg/m2) to avoid anastomosis-related complications and potentially improve the surgical outcome of esophagectomy for esophageal cancer in such patients.” (Page 16 & 17, lines 279–293)

 

To Reviewer #2

Reviewer’s comment 1

A. The results state, that retrosternal approach might be associated to higher anastomotic leakage in higher BMI patients due to a longer conduit.

B. Clavicular or sternal resection is often needed to avoid postoperative dysphagia in patients with retrosternal approach. This is congruent to our experience. Therefore our clinic has set the posterior mediastinal approach as the standard procedure.

 In which case would you still recommend retrosternal approach and therefore prove relevance of these findings?

Response to comment 1

In Japan, the PM and RS routes are the standard reconstruction routes following radical esophagectomy for cancer in 40.2% and 40.1% of cases, respectively. The selection of the reconstruction route depends on the surgeon and institute, and the surgical outcomes of esophagectomy do not significantly differ between both routes. In our department, the RS route is the standard reconstruction route employed in cases where reconstruction using the PM route may result in serious complications, such as esophagobronchial fistula, and the manubrium or clavicular resection is not performed. The present study corroborated the results of previous studies in Japan by showing that surgical outcomes of esophagectomy do not differ between both routes. The results of our study suggest that the PM route should be selected to avoid anastomosis-related complications when treating obese patients with a high perioperative risk. According to your suggestion, we have added the following sentence to the Discussion: “Thus, even if the standard route for reconstruction is RS, we recommend using the PM route when treating obese patients (BMI >25 kg/m2) to avoid anastomosis-related complications and potentially improve the surgical outcome of esophagectomy for esophageal cancer in such patients.” (Pages 16 & 17, lines 290–293). 

Reviewer’s comment 2

In your manuscript you mentioned that higher BMI is associated to longer routes for RS routes. Fig 3 shows some kind of visualization. Nevertheless, for clinical use, a consequence of these findings could be stated. Is there possibly a cut-off value for significant lengthening of the route? Is there a recommendation for which approach to prioritize or avoid with a certain BMI?

Response to comment 2

Thank you for your pertinent comment. In this study, longer RS routes were observed in patients with a BMI of 19.6 kg/m2 or above (area under the curve, 0.8). This cut-off value is described in the Results section (Page 10, line 186). According to your suggestion, we have added the following sentence to the Discussion section: “In this study, five overweight cases had BMIs >25 kg/m2, including one with BMI >30 kg/m2; in all these cases, the RS route was longer than the PM route. Thus, even if the standard route for reconstruction is RS, we recommend using the PM route when treating obese patients (BMI >25 kg/m2) to avoid anastomosis-related complications and potentially improve the surgical outcome of esophagectomy for esophageal cancer in such patients.” (Pages 16 & 17, lines 288–293).

Reviewer’s comment 3

The discussion lacks a conclusion for clinicians towards which consequence your findings might have (literature states that anastomotic leakage rates overall are not higher in the retrosternal approach than in the posterior mediastinal approach, as you stated earlier in your manuscript).

Response to comment 3

Thank you for your suggestion. This study found that the RS route was longer in patients with a high BMI but did not determine the relationships between reconstruction length and surgical outcomes of esophagectomy in obese patients. Although no significant difference is observed between surgical outcomes of esophagectomy using the two routes, there is room to further investigate how reconstruction length is associated with surgical outcome of esophagectomy in patients with a high BMI. The results of this study indicate that practicing esophageal surgeons could consider adding the Kocher maneuver and creating a long gastric conduit when treating obese patients. Furthermore, employing the PM instead of the RS route may improve the outcomes of esophagectomy for esophageal cancer in such patients. According to your suggestion, we have added the following sentence to the Discussion section: “Clinicians could consider using the Kocher maneuver and creating a long gastric conduit when treating obese patients.” (Page 13, lines 222–223)

---

## [Decision Letter · Decision Letter 1]

5 Dec 2022

PONE-D-21-33230R1An individual’s physique affects the length of the reconstruction route via the retrosternal approachPLOS ONE

Dear Dr. Lee,

Thank you for submitting your manuscript to PLOS ONE. After careful consideration, we feel that it has merit but does not fully meet PLOS ONE’s publication criteria as it currently stands. Therefore, we invite you to submit a revised version of the manuscript that addresses the points raised during the review process.

We look forward to receiving your revised manuscript.

Kind regards,

Callam Davidson

Editorial Office

PLOS ONE

Journal Requirements:

Your study is observational and therefore causality cannot be inferred. Please update your title to reflect this (replace ‘affects’ with ‘is associated with’) and check throughout the manuscript for similar instances (I found examples in both the abstract and conclusions).

Reviewers' comments:

Reviewer's Responses to Questions

**Comments to the Author**

1. If the authors have adequately addressed your comments raised in a previous round of review and you feel that this manuscript is now acceptable for publication, you may indicate that here to bypass the “Comments to the Author” section, enter your conflict of interest statement in the “Confidential to Editor” section, and submit your "Accept" recommendation.

Reviewer #2: All comments have been addressed

Reviewer #3: All comments have been addressed

2. Is the manuscript technically sound, and do the data support the conclusions?

Reviewer #2: Yes

Reviewer #3: Partly

3. Has the statistical analysis been performed appropriately and rigorously? 

Reviewer #2: Yes

Reviewer #3: Yes

4. Have the authors made all data underlying the findings in their manuscript fully available?

Reviewer #2: Yes

Reviewer #3: Yes

5. Is the manuscript presented in an intelligible fashion and written in standard English?

Reviewer #2: Yes

Reviewer #3: Yes

6. Review Comments to the Author

Reviewer #2: Thank you very much for addressing all comments and adding details to the discussion. Congratulations on your acceptance!

Reviewer #3: Honda et al. used 3DCT to measure the distance between the posterior mediastinal route and the retrosternal route during esophagectomy, and reported that high BMI was associated with the distance in the retrosternal route. Your manuscript is well written and methodology is adequate. Reviewer has a few questions and comments.

1. The authors concluded that choosing the posterior mediastinum route reduces anastomosis-related complications in obese patients with long distance of the retrosternal reconstruction route. However, the relationship between the length of the reconstruction route and the incidence of anastomotic leakage was not shown in this paper.

2. As for the similar question, is there a difference in the incidence of suture failure between the RS ≥ ES group and the RS < ES group?

3. Please add the references of Collard and triangulating stapled anastomosis. (line 79-80)

7. PLOS authors have the option to publish the peer review history of their article (what does this mean?). If published, this will include your full peer review and any attached files.

Reviewer #2: No

Reviewer #3: No

---

## [Author Response · Author response to Decision Letter 1]

30 Dec 2022

Responses to Editor’s and Reviewers’ Comments

To Editor

Editor’s comment

Your study is observational and therefore causality cannot be inferred. Please update your title to reflect this (replace ‘affects’ with ‘is associated with’) and check throughout the manuscript for similar instances (I found examples in both the abstract and conclusions).

Response to comment

Thank you for your kind suggestion. We have checked throughout the manuscript for similar instances. We have replaced “affect” with “is associated with” in the title and body of the manuscript according to your suggestion (Page 1, lines 1 & Page 1, lines 15 & Page 2, lines 18 & Page 2, lines 21 & Page 3, lines 21 & Page 5, lines 5).

To Reviewer #3

Reviewer’s comment 1

The authors concluded that choosing the posterior mediastinum route reduces anastomosis-related complications in obese patients with long distance of the retrosternal reconstruction route. However, the relationship between the length of the reconstruction route and the incidence of anastomotic leakage was not shown in this paper.

Response to comment 1

Thank you for your pertinent comment. In the RS reconstructions (n=40), the RS ≥ ES group had 30 cases and the RS < ES group had 10 cases; anastomosis-related complications occurred in six cases and one case in the two groups, respectively, which were not significantly different (p=0.66). Our series did not reveal a relationship between the length of the reconstruction route and anastomotic leakage because of the small number of cases. Furthermore, five overweight cases had BMIs >25 kg/m2; in all these cases, the RS route was longer than the PM route, and no anastomosis-related complications were observed. Our study revealed no relationship between BMI and anastomotic leakage; however, the study findings revealed that the reconstruction length was related to the individual’s physique. Further studies exploring the relationships between reconstruction lengths, individual’s physique, and outcomes of esophagectomies are needed to confirm these findings. According to your suggestion, we have added the following sentence to the Results and Discussion sections: “The RS ≥ ES group included 30 cases, whereas the RS < ES group included 10 cases; meanwhile, the anastomosis-related complications in each group comprised six cases and one case, respectively, in which there was no significant difference (p=0.66)” (Page 10, lines 7-10). “Although our series did not reveal a relationship between the length of the reconstruction route and anastomotic leakage and did not reveal a relationship between the BMI and anastomotic leakage” (Page 16, lines 6-7).

Reviewer’s comment 2

As for the similar question, is there a difference in the incidence of suture failure between the RS ≥ ES group and the RS < ES group?

Response to comment 2

Thank you for your comment. As noted in the response to comment 1, there was no difference in the incidence of suture failure between the RS ≥ ES and RS < ES groups. According to your suggestion, we have added the following sentence to the Results section: “The RS ≥ ES group included 30 cases, whereas the RS < ES group included 10 cases; meanwhile, the anastomosis-related complications in each group comprised six cases and one case, respectively, in which there was no significant difference (p=0.66)” (Page 10, lines 7-10).

Reviewer’s comment 3

Please add the references of Collard and triangulating stapled anastomosis. (line 79-80)

Response to comment 3

Thank you for your advice. We have added the following references for Collard and triangulating stapled anastomosis according to your suggestion: 

22. Collard JM, Romagnoli R, Goncette L, Otte JB, Kestens PJ. Terminalized semimechanical side-to-side suture technique for cervical esophagogastrostomy. Ann Thorac Surg. 1998;65: 814-817.

23. Orringer MB, Marshall B, Iannettoni MD. Eliminating the cervical esophagogastric anastomotic leak with a side‐to‐side stapled anastomosis. J Thorac Cardiovasc Surg. 2000;119: 277-288.

24. Furukawa Y, Hanyu N, Hirai K, Ushigome T, Kawasaki N, Toyama Y, et al. Usefulness of automatic triangular anastomosis for esophageal cancer surgery using a linear stapler (TA-30). Ann Thorac Cardiovasc Surg. 2005;11: 80-86.

---

## [Decision Letter · Decision Letter 2]

20 Mar 2023

An individual’s physique is associated with the length of the reconstruction route via the retrosternal approach

PONE-D-21-33230R2

Dear Dr. Lee,

We’re pleased to inform you that your manuscript has been judged scientifically suitable for publication and will be formally accepted for publication once it meets all outstanding technical requirements.

Kind regards,

Mohamad Khair Abou Chaar

Academic Editor

PLOS ONE

Additional Editor Comments (optional):

Dear authors,

I want to congratulate you on the work you presented and thank you for making the modifications requested by our reviewers. I have no further comments and I believe that the manuscript is fit for publication.

Thank you,

Mohamad Khair Abou Chaar, MD

Reviewers' comments:

Reviewer's Responses to Questions

**Comments to the Author**

1. If the authors have adequately addressed your comments raised in a previous round of review and you feel that this manuscript is now acceptable for publication, you may indicate that here to bypass the “Comments to the Author” section, enter your conflict of interest statement in the “Confidential to Editor” section, and submit your "Accept" recommendation.

Reviewer #3: All comments have been addressed

2. Is the manuscript technically sound, and do the data support the conclusions?

Reviewer #3: Yes

3. Has the statistical analysis been performed appropriately and rigorously? 

Reviewer #3: Yes

4. Have the authors made all data underlying the findings in their manuscript fully available?

Reviewer #3: Yes

5. Is the manuscript presented in an intelligible fashion and written in standard English?

Reviewer #3: Yes

6. Review Comments to the Author

Reviewer #3: (No Response)

7. PLOS authors have the option to publish the peer review history of their article (what does this mean?). If published, this will include your full peer review and any attached files.

Reviewer #3: No

---

## [Editor Report · Acceptance letter]

23 Mar 2023

PONE-D-21-33230R2 

An individual’s physique is associated with the length of the reconstruction route via the retrosternal approach 

Dear Dr. Lee:

I'm pleased to inform you that your manuscript has been deemed suitable for publication in PLOS ONE. Congratulations! Your manuscript is now with our production department. 

Kind regards, 

on behalf of

Dr. Mohamad Khair Abou Chaar 

Academic Editor

PLOS ONE